# Pleasant Stroke Touch on Human Back by a Human and a Robot

**DOI:** 10.3390/s23031136

**Published:** 2023-01-19

**Authors:** Tomoki Ishikura, Yuki Kitamura, Wataru Sato, Jun Takamatsu, Akishige Yuguchi, Sung-Gwi Cho, Ming Ding, Sakiko Yoshikawa, Tsukasa Ogasawara

**Affiliations:** 1Division of Information Science, Nara Institute of Science and Technology, Ikoma 630-0192, Japan; 2Psychological Process Research Team, Guardian Robot Project, RIKEN, Seika-cho, Soraku-gun, Kyoto 619-0228, Japan; 3Knowledge Acquisition and Dialogue Research Team, Guardian Robot Project, RIKEN, Seika-cho, Soraku-gun, Kyoto 619-0228, Japan; 4Division of Electronic Engineering, School of Science and Engineering, Tokyo Denki University, Hatoyama-machi, Hiki-gun, Saitama 350-0394, Japan; 5Institutes of Innovation for Future Society, Nagoya University, Nagoya 464-8601, Japan; 6Faculty of Art and Design, Kyoto University of the Arts, Kyoto 606-8271, Japan

**Keywords:** touch by a robot, social interaction, subjective and physiological measure

## Abstract

Pleasant touching is an important aspect of social interactions that is widely used as a caregiving technique. To address the problems resulting from a lack of available human caregivers, previous research has attempted to develop robots that can perform this kind of pleasant touch. However, it remains unclear whether robots can provide such a pleasant touch in a manner similar to humans. To investigate this issue, we compared the effect of the speed of gentle strokes on the back between human and robot agents on the emotional responses of human participants (*n* = 28). A robot or a human stroked on the participants’ back at two different speeds (i.e., 2.6 and 8.5 cm/s). The participants’ subjective (valence and arousal ratings) and physiological (facial electromyography (EMG) recorded from the corrugator supercilii and zygomatic major muscles and skin conductance response) emotional reactions were measured. The subjective ratings demonstrated that the speed of 8.5 cm/s was more pleasant and arousing than the speed of 2.6 cm/s for both human and robot strokes. The corrugator supercilii EMG showed that the speed of 8.5 cm/s resulted in reduced activity in response to both human and robot strokes. These results demonstrate similar speed-dependent modulations of stroke on subjective and physiological positive emotional responses across human and robot agents and suggest that robots can provide a pleasant touch similar to that of humans.

## 1. Introduction

Physical touch with others is an important aspect of social interactions that is often used in nonverbal communication. Skillful or purposeful touch is important for several caregiving techniques, including *tactile massage* [1] and *humanitude* [2]. It has been reported in several clinical studies that stroking a patient’s back in tactile massage had positive effects, e.g., reductions in fatigue and pain and improvement in the quality of sleep [3]. A touching skill in *humanitude* care can reduce the frequency of the behavioral and psychological symptoms associated with dementia for elderly people, compared to standard care [4].

Experimental psycho-physiological studies have investigated the effect of touch stroke speed and subjective/physiological emotional responses induced by touch. In this study, we focus on the relationship between stroke speed and emotional responses, following this research stream. Such testing was conducted with a brush [5,6,7]. Ree et al. [6] provided brush strokes to an arm at speeds of 3 and 0.3 cm/s and measured participants’ subjective ratings of emotional valence and intensity, and they measured the physiological responses of facial electromyography (EMG) from the corrugator supercilii (i.e., brow lowering) and zygomatic major (i.e., lip corner pulling) muscles and the skin conductance level. The results showed that 3 cm/s touch induced higher valence and intensity ratings than with 0.3 cm/s touch. Corresponding to these subjective ratings [8,9,10], touch at the faster speed also induced weaker activity in the corrugator supercilii EMG. These subjective and physiological responses, which have been replicated in other studies with some differences [5,7], suggest that the effect of touch can be measured by subjective and physiological emotional responses. The effect of speed is generally interpreted in that touch at a speed of 1 to 10 cm/s will appropriately stimulate the C-tactile (CT) fibers that respond to slow and gentle stroking touch and induce emotional responses. A study further suggested that emotional responses could differ across the 1 to 10 cm/s CT-optimal range, through the patterns could differ across the sites on the body [11]. These data suggest that human’s stroke touch can induce positive subjective and physiological emotional responses, which could differ depending on the stroking speed.

However, it remains unknown whether agents other than humans, specifically robots, could provide pleasant touch that functions similar to humans’ touch. Robots capable of providing such a pleasant touch could help alleviate problems by a lack of caregivers and thereby improve the quality of caregiving generally [12,13]. Some previous studies have shown that touch by a robot can provide a positive effect for humans [14,15]. However, to the best of our knowledge, no study has directly compared the touch provided by humans with the touch provided by robot agents, and no study has shown whether imitation learning [16] (i.e., mimicking the human’s performance) can be used to realize the pleasant touch of a robot. Hence, we hypothesized that smartly developed robotic touch could induce a similar emotional effect to human touch. In other words, the robotic stroke would induce the same changing patterns, which depend on the speed of the stroke, resulting in similar changes to human participants’ subjective and physiological emotional responses if the participants do not know whether the agent stroking their back is a human or a robot.

To verify our hypothesis, we developed a robot agent that could provide gentle strokes on participants’ back (Figure 1). In this study, we selected the back as the target location of touching because this is suitable for designing our experiment (i.e., a paticipant cannot directly see the stroking agent) and because stroking a patient’s back is commonly used in tactile massage. The designed robot could provide strokes at slower 2.6 cm/s and faster 8.5 cm/s speeds that could stimulate CT fibers. These two speeds are easy to use for human safety, and such CT-optimal speeds may have different psychological effects, as Ackerley et al. [11] pointed out. We investigated the emotional effects elicited by these two speeds of strokes by two agents (a human and a robot) using subjective and physiological responses. To ensure the participants were not biased based on their preconceptions of robots and humans, participants were instructed that only a robot would stroke their back, but a robot or a human touched the participants’ back without the participants observation. Debriefing interviews confirmed that none of participants knew that a human also provided strokes.

For subjective responses, we assessed the valence and arousal ratings using an affect grid, wherein the valence and arousal represented the qualitative and energetic components of emotions, respectively [17]. For physiological responses, we measured the facial EMG from the corrugator supercilii and zygomatic major muscles and the skin conductance response (SCR). The former two and the latter measure reflected emotional valence and arousal responses, respectively [8,9,10]. Here, we predicted that the robot and human agents would elicit similar subjective and physiological emotional response patterns, depending on the speed conditions, although we had no prediction regarding the direction of differences across speeds, due to a lack of data in the literature.

## 2. Touch by a Robot

Under the assumption that a human-like pleasant stroke is also a pleasant stroke for a robot, we developed a human-mimetic robotic hand, wherein the average temperature, size, and softness of the human hand were imitated. We used a corroborative robot manipulator, *UR3e*, Universal Robots (https://www.universal-robots.com/e-series, accessed on 23 October 2022), to move the hand to stroke the back.

### 2.1. Human-Mimetic Robotic Hand

In our previous research [18], we found that the following two elements are important to creating a robot hand for pleasant touch: (1) the mechanism should be designed to fit the stroking surface; (2) the temperature should be warm, similar to a human hand. We developed a human-mimetic robotic hand, which satisfies the following three aspects:Flexibility to naturally conform to the surface being touched;Similar softness and stiffness as a human hand;Similar warmness as a human hand.

As shown in Figure 2, the developed human-mimetic robotic hand consists of bones, joints, heaters, and material mimetic body-tissue that provides softness, while the inside bones provide stiffness similar to a human hand. The bones were made with 3D printed materials. The shape was a human skeleton model downloaded from *STLFinder* (https://www.stlfinder.com, accessed on 23 October 2022). As shown in Figure 3, between the bones, the torsion springs (0.3 Nmm/deg) were used as the joints to make them flexible. Moreover, to fabricate heaters, we wound nichrome wires around the bones, attached film-type thermistor sensors, and covered them with aluminum foils to uniformly warm the human-mimetic robotic hand. We used a type of soft silicon (HITOHADA Gel, EXSEAL Co., Ltd., Gifu, Japan, https://www.exseal.co.jp/en/products/#product_05, accessed on 23 October 2022), which has softness similar to human skin to cover the bones, joints, and heaters. Finally, we used natural rubber gloves to cover and protect the silicon.

We heated nichrome wires to control the temperature of the human-mimetic robotic hand with temperature sensors (film-type thermistors). By dividing the hand into six parts, the burden of each heater was reduced. The six parts were a thumb, an index finger, a middle finger, a ring finger, a pinky finger, and a palm. Thus, we equipped the developed hand with six sets of the heater and the sensor. The temperatures in the six parts of the hand were independently controlled. A PID controller controlled the temperature of each heater. In the PID controller, the desired value was set so that the surface of the robotic hand becomes 35 °C.

### 2.2. Generation of Stroke Motion

We attached the proposed human-mimetic robotic hand to the end of the robot arm to stroke the back of a human. Based on the knowledge of tactile care [19], to generate gentle stroke motions on the back of a human using the robot arm, we tried to keep constant pressure. To do so, we adopted the impedance control. The robot arm used in this study had the impedance controller provided by the manufacturer, and we used it as is to unify the robot control for each subject as much as possible; we believe that this control is built inside of the robot control loop and is more reliable than the controller that is built outside of the control loop. We set the target force to 3 N, following the suggestion in [15].

Figure 4 shows the flow to stroke the back of a subject using the robot arm. First, the robot arm moved toward the back of a subject on 1 cm/s, and the robot arm kept to press the middle back of the subject. For removing a physiological effect from the initial contact, the robot kept pressing for 10 s to wait for a subject to stay calm. Next, the robot arm moved down 15 cm and returned the touching position. The robot arm repeated the same motion for around 20 s. After the robot arm finished stroking, the robot arm took to leave the back of the subject at 30 cm/s.

When stroking the back, the robot arm moved the human-mimetic robotic hand in a constant acceleration of 0.2 cm/s^2^, until reaching the predefined maximum speed. Then the robot arm maintained its speed. When approaching the goal position, the robot arm decelerated the robotic hand at 0.2 cm/s^2^. In the slower stroke, we set the maximum speed to 3 cm/s. The velocity profile was a trapezoid with a height of 3 cm/s. The average speed was 2.6 cm/s. In the faster stroke, we generated the motion as fast as possible in the range of the acceleration. We set the maximum speed to 30 cm/s, but the maximum speed was not reached because of the short distance of the movements on the back. The velocity profile was a triangle with a height of 17 cm/s. The average speed was 8.5 cm/s.

## 3. Experimental Setup

### 3.1. Participants

Thirty healthy males (age: 23.1 ± 2.9) were recruited for the experiment. The main argument of this study is to show whether robots can provide pleasant touch strokes in a manner similar to humans. As variable touch effects across genders have been reported [20], we recruited only male participants. Additionally, we attempted to clarify this argument by reducing the complicated individual factors by limiting the age range. The number of participants was determined based on a priori power analysis using the G*Power software ver. 3.1.9.2 [21]. Assuming an α level of 0.05, a power of 0.80, repeated-measures correlation of 0.6, and medium effect size (*f* = 0.25), the results indicated that 28 participants were required for the planned analyses. Each participant first answered the questionnaire, which asked about their age, height, and weight. We asked the participant to wear earmuffs to reduce the noise from the robot and experimenters during the experiment. We also asked all participants to wear the same patient wear (SG-1441, Nagaileben co., Ltd., Tokyo, Japan https://www.nagaileben.co.jp/Webcatalog_Naway2023/#page=302, accessed on 23 October 2022) over their inner wear to control the stimuli to the back. We explained the procedure of the experiment and obtained informed consent by having the participants sign the consent. Due to the experimental procedure, we did not inform participants that two different agents would stroke their back; however, we notified the subjects of the existence of the two agents after the experiment concluded. This study was approved by the ethics committee of Nara Institute of Science and Technology, Japan and was conducted according to institutional ethical provisions and the Declaration of Helsinki.

### 3.2. Experimental Design

#### 3.2.1. Conditions

To assess our aforementioned hypothesis, we carefully designed the experimental conditions:We prepared two agents to provide strokes: a human male volunteer and a robot.We prepared two types of strokes: faster (8.5 cm/s) and slower (2.8 cm/s) strokes.

Thus, there were four conditions (two agents × two speeds). Note that previous research [5,6,7] has evaluated the emotional difference imparted by different stroke speeds similar to the ones used here.

#### 3.2.2. Key Points for Experimental Design

We did not inform the participant that there were two agents, because we wanted the participant to evaluate the stimuli of the strokes alone. This procedure was designed to prevent a situation in which the participant perceived the existence of the human agent and evaluated strokes by distinguishing the difference between the human and robot agents. We considered the following four aspects from which a participant might perceive the existence of the human agent: (1) visual information; (2) the noise from a robot; (3) variation in the speed of human strokes in all trials; and (4) variation in the force of human and robot strokes in all trials. Based on our post-experiment evaluation, none of participants were able to perceive the existence of a human agent in the experiment.

To eliminate the potential bias of the visual cues, we chose the back of the subject as the target for stroke. Thus, the participant was unable to see the robot stroke directly. As described above, stroking the back is very common in *tactile massage* [1]. Not seeing a robot directly was useful for avoiding any potential effects of the robot’s appearance on the participant’s evaluation. For example, Chen et al. [15] reported that the mechanical appearance and behavior of a robot affected the participant’s emotions and altered their subjective evaluation. Although the human agent had to be near the participant to provide the stroke, we notified the participant that the human was always near the participant as a precaution to stop the robot’s movement in the unlikely case of emergency.

To eliminate to potential bias of auditory cues, as shown in Figure 4 and Figure 5, the robot moved even if a human stroked the back. Thus, the noise from the robot was consistent across all conditions. In addition, the participant wore earmuffs to reduce the noise.

Regarding the third point (i.e., the variation in the speed of human strokes in all trials), although it is easy for the robot to repeat the same movements over subsequent trials, this is not the same for humans. Thus, the robot guided the speed by showing the movement to the human.

Lastly, regarding variations in force, the robot used a controlled 3 N of force by using the impedance control provided by the robot manufacturer. The human, alternatively, practiced to become familiar and consistent with applying 3 N of force by training with a force plate before the experiment.

### 3.3. Procedure

The experiment was conducted in a soundproof room. To enable air circulation, the door was kept open. Figure 6 shows the procedure of the experiment. First, to allow the participant to get used to a robot stroke, the participant experienced robot strokes without evaluation. Then, we repeated the following four steps (each referred to as one trial) forty times: (1) One of the four conditions was chosen in a pseudo-random manner. (2) The chosen agent maintained touch with the back of the participant for ten seconds to remove the physiological effect caused by the moment of contact. (3) The chosen type of stroke was provided for 20 s. During the stroke, the physiological signals of the participant were measured. We unified the amount of the stimuli from different types of strokes by unifying the duration. (4) The participant subjectively evaluated the stroke for 30 s. Note that each condition was conducted ten times. After forty trials, the participant answered the free-description questionnaire to subjectively evaluate the entire experiment.

### 3.4. Evaluation Indices

As subjective indices, we used the affect grid [17], which graphically represented the two dimensions of valence and arousal using a 9-point scale. The participants were instructed to rate their subjective emotional experience after receiving touch by indicating the coordinates on the affect grid. For physiological evaluations, we used facial EMG recorded from the corrugator supercilii and zygomatic major muscles and SCR. We followed the physiological evaluation method proposed by Mayo et al. [5] and Ree et al. [6].

To measure the facial EMG, we followed the guideline proposed by Fridlund et al. [22] Ag/AgCl electrode pads were attached directly above the subject’s left corrugator supercilii muscle and left zygomatic major muscle. The facial EMG signals were amplified by EMG-025. Finally, the amplified signals were sampled at 1000 Hz by the PowerLab 16/35 data acquisition system and LabChart Pro v8.0 software.

To measure the SCR, the Ag/AgCl electrode pads were attached around the middle phalanges (MP) of the index and middle finger. By applying a constant voltage of 0.5 V between the fingers, the SCR was measured using a Model 2701 BioDerm Skin Conductance Meter (UFI, Morro Bay, CA, USA). The measured data were sampled at 1000 Hz using the same equipment that measured the facial EMG.

### 3.5. Data Analysis

Based on related research [10], the facial EMG data were analyzed according to the following four steps: (1) EMG data were processed by full-wave rectification; (2) EMG data for the one second just before stroking began were averaged, and the averaged value was set as the baseline EMG data; (3) the data during stroking were divided into one second bins, and the average EMG data for each bin was calculated; (4) the difference between the average EMG data in each division and the baseline EMG data was calculated. To remove the individual differences in the data, we normalized the data in Step 4 for all trials to adjust the average and the standard deviation of the normalized data to zero and one, respectively.

Based on related research [23], the SCR data were analyzed in the following three steps: (1) SCR data for one second just before stroking were averaged, and the averaged value was set to the baseline SCR data; (2) the maximum SCR data during stroking was obtained; and (3) the difference between the maximum and baseline SCR data was calculated. To remove individual differences, the differences for all trials were normalized, similar to the process for the facial EMG data.

Because the data of two participants were not appropriately obtained, due to the accidental malfunction of the physiological measurement, we analyzed the data obtained from the 40 trials of the remaining 28 subjects. To assess the aforementioned hypothesis, we applied within-subjects analysis of variance (ANOVA) to the data. In the analysis, we set the subjective evaluations (valence and arousal) from the affect grid and physiological signals (facial EMG and SCR) as the dependent variables and the two types of agents and two speed variations as the independent variables.

## 4. Results

Figure 7 and Figure 8 show the subjective and physiological responses to touch strokes at two speeds using two agents. Physiological data were analyzed after standardization within individuals to satisfy the normal distribution assumption.

### 4.1. Subjective Evaluations

For valence ratings, the ANOVA revealed that the main effects of the speed and agent, as well as the interaction between these factors, were significant (*F*s(1, 27) = 13.41, 31.53, and 5.25, *p*s = 0.001, 0.000, and 0.030, respectively). The main effects of the speed and agent indicated that touch at the faster speed by humans induced more positive responses. Follow-up analyses of the interaction showed that the simple main effects of speed were significant for both the human and robot agents (*F*s(1, 54) = 18.5 and 4.96, *p*s = 0.000, and 0.030, respectively), indicating that both agents elicited more positive responses at the faster than the slower speed.

For arousal, the ANOVA revealed a significant main effect of speed, indicating higher arousal for the faster speed than the slower speed, as well as a significant interaction between speed and agent (*F*s(1, 27) = 19.97 and 7.08, *p*s = 0.000 and 0.013, respectively). The main effect of the agent was not significant (*F*(1, 27) = 0.10, *p* = 0.754). Follow-up analyses of the interaction showed that the simple main effects of speed were significant for both the human and robot agents (*F*s(1, 54) = 25.39 and 12.48, *p*s = 0.000 and 0.001, respectively), indicating that both agents induced higher arousal at the faster, rather than slower, speed.

### 4.2. Physiological Evaluations

For the facial EMG recorded from the corrugator supercilii muscle, the ANOVA revealed that only the main effect of speed was significant (*F*(1, 27) = 6.32, *p* = 0.018), indicating that touch at the faster speed induced a weaker corrugator supercilii muscle activity, which is pleasant. The main effect of agent and the interaction between speed and agent were not significant (*F*s(1, 27) = 0.10 and 0.05, *p*s = 0.758 and 0.829, respectively).

For the zygomatic major muscle EMG, none of the main effects of speed and agent and their interaction were significant (*F*s(1, 27) = 0.19, 0.08, and 1.00, *p*s = 0.665, 0.785, and 0.325, respectively).

For SCR, the ANOVA revealed a non-significant trend for the main effect of speed and the interaction between the speed and agent (*F*s(1, 27) = 3.37 and 2.99, *p*s = 0.077, and 0.095, respectively). The main effect of the agent was not significant (*F*(1, 27) = 0.10, *p* = 0.758).

## 5. Discussion

### Robot Motion

Our results of the subjective valence ratings and corrugator supercilii EMG responses supported our hypothesis that robot and human strokes would induce the same changes in the patterns of emotional responses, depending on the speed parameters. Specifically, the strokes to the back were more pleasant at the faster (i.e., 8.5 cm/s), rather than the slower (i.e., 2.6 cm/s), speed when provided by both the robot and human agents. These results, showing an elicitation of positive emotional responses by strokes at these speeds, are in line with the previous findings that found that strokes to arms with brushes at speeds stimulating CT fibers (i.e., the 1 to 10 cm/s) induced positive subjective and physiological emotional responses [5,6,7]. Our result of different responses across different speed conditions is also consistent with the previous finding that emotional responses can differ across speeds in a site-specific manner [11], although the touch on the back was not tested in previous research. The current results extend these findings and demonstrate that the touch on the back by human and robot agents induces speed-dependent positive emotional responses in the same manner.

The results of the subjective arousal ratings also showed that, for both human and robot strokes, faster speed strokes elicited higher arousal responses than did slow strokes, further supporting our hypothesis that these two agents can induce the same speed-dependent patterns in emotional responses. In contrast, the SCR did not show evident differences across the speed condition for both agents. The result is in line with the results of a previous study reporting that SCR did not show significant differences between optimal and non-optimal conditions to stimulate CT fibers for strokes made to the arms using a brush [6], suggesting that the effect of stroke on autonomic nervous system activity may be weak, relative to the effect on facial EMG signals.

Our conclusions regarding the arousal responses are in-line with that of previous research. Pawling et al. [7] showed that slow strokes significantly reduced heart rate (which corresponds to a low arousal response) more than did fast strokes, because slow strokes activate the parasympathetic nerve more. Zamani et al. [24] also showed that fast tapping to both a forearm and a shoulder elicited high arousal responses.

To the best of our knowledge, this is the first study evaluating the effect of strokes by not only a human, but also a robot, with both valence and arousal indices using subjective evaluation and physiological signals. Our results support that a robot can act as an agent capable of inducing pleasant emotional responses by touch in a manner similar to humans. The results have practical significance, in that future robots based on this design could act as caregivers that provide patients with skillful touches as part of therapeutic or health interventions. To accomplish this objective, future experiments should be conducted under more practical scenarios, such as testing gentle strokes such as tactile massage.

The developed robot system had two limitations. First, the strokes by the current system were viewed as slightly worse, with respect to valence, than the human strokes provided at the same speed. In the current system, the normal direction of the robot hand was maintained in the horizontal orientation during stroking. As the result, the touching area was somewhat smaller than that of the human stroke. In the free-description questionnaire, several participants described that the smaller touching areas reduced their enjoyment of the interaction. We believe that the system could be improved by adding the mechanisms to allow the robot hand to cover a greater area and by generating a trajectory to allow the hand to follow the natural curve of the back. The second limitation is that we did not measure the activities of the CT fibers during stroking. If we had measured the fibers directly, we would have been able to optimize the pleasant strokes by finding the stroke patterns that activate the CT fibers maximally.

## 6. Conclusions

In this study, we compared the effect of stroke speed for a hand stroking the human back between human and robot agents and evaluated the human’s subjective and physiological emotional responses. We found the same changing patterns, depending on the speed parameters, across the human and robot agents in both the valence ratings and the corrugator supercilii EMG activity. These results suggest that robots can provide pleasant touch in a manner similar to that of humans.

In this study, we investigated the relationship between the stroke speed and the pleasantness of the stroke. Note that there are other possible factors that determine pleasantness, such as pressure. In the future, we would like to investigate other factors related to pleasantness. Although the experiments in this paper were conducted in a situation where the robot was not directly seen, it is also important to examine the psychological effects of seeing the robot stroking the back. Another future direction is to clarify the difference in emotional responses of individual factors, such as younger/elder participants and male/female participants.

## Figures and Tables

**Figure 1 sensors-23-01136-f001:**
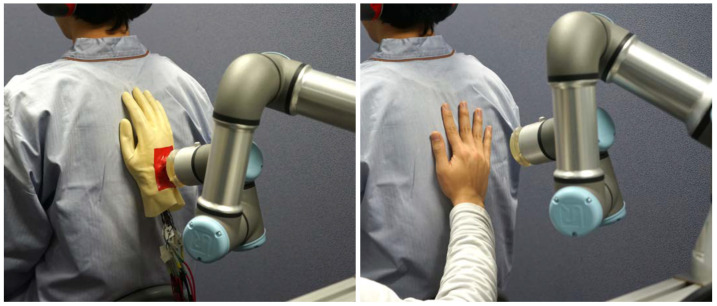
Comparative experimental conditions for the robot stroke motion (**left**) and the human stroke motion (**right**).

**Figure 2 sensors-23-01136-f002:**
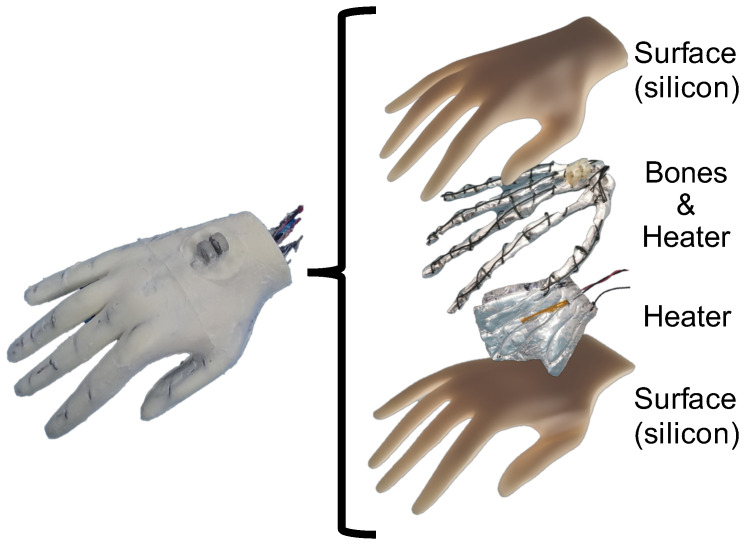
The structure of the developed human-mimetic robotic hand.

**Figure 3 sensors-23-01136-f003:**
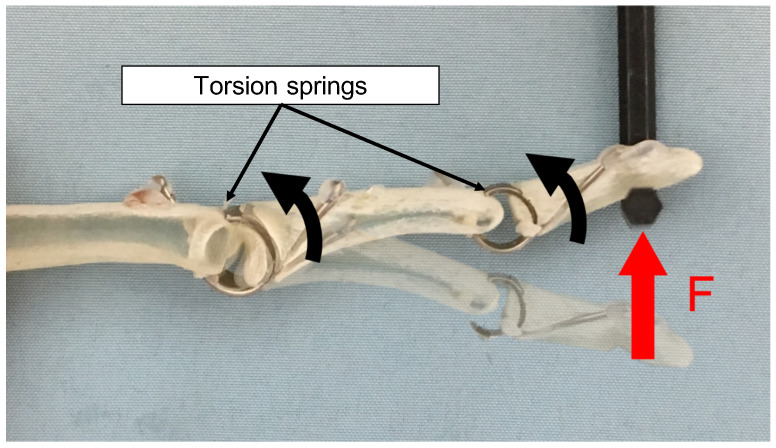
Transformation of the thumb with two passive joints.

**Figure 4 sensors-23-01136-f004:**
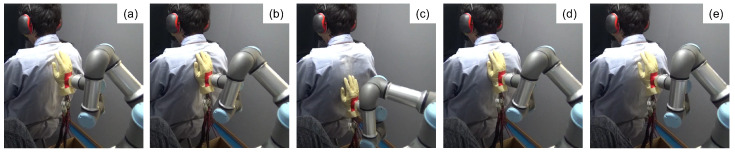
The flow of the robot stroke motion. (**a**) Move towards the back. (**b**) Contact the back and keep pressure. (**c**,**d**) Stroke the back and return. (**e**) Leave the back.

**Figure 5 sensors-23-01136-f005:**
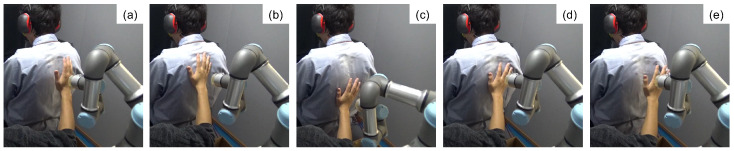
Experimental scene with the human agent. (**a**) Move towards the back. (**b**) Contact the back and keep pressure. (**c**,**d**) Stroke up and down the back. (**e**) Leave the back.

**Figure 6 sensors-23-01136-f006:**
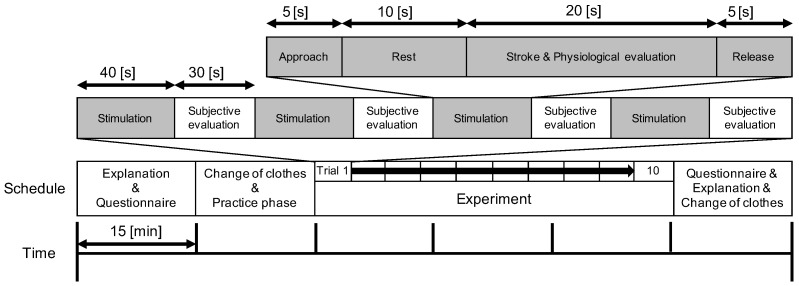
The experimental flow.

**Figure 7 sensors-23-01136-f007:**
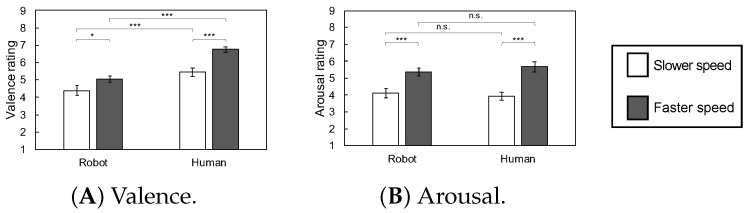
The mean with SE of the subjective evaluation. (***: *p* < 0.001, *: *p* < 0.05, n.s.: *p* > 0.10).

**Figure 8 sensors-23-01136-f008:**
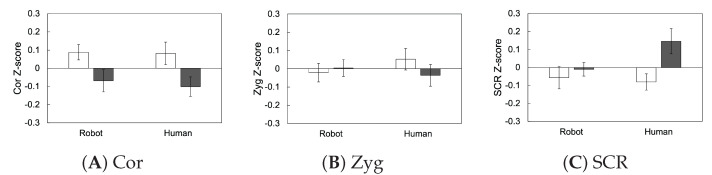
The mean with SE of the physiological evaluation. Cor: corrugator supercilii EMG; Zyg: zygomatic major EMG; SCR: skin conductance response.

## Data Availability

Not applicable.

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
