# Peer review of "Pleasant Stroke Touch on Human Back by a Human and a Robot"

_sensors, 2023, doi:10.3390/s23031136_

Round 1

Reviewer 1 Report

This manuscript developed a human-mimetic robotic hand to investigated the emotional effects between the human hand and the robotic hand. But the manuscript only focuses on the moving speed of the hands stroke, some other factors, i.e., pressure, should better be consider in the researching. For example, when a human hand strokes on the back, each finger and palm actually provide different pressure. Such that, the controlling algorithms of robotic hand and the fingers should be designed and verified. 

At the same time, only thirty healthy males (age: 23.1±2.9) are recruited for the experiment, which means that the experiment havent covered enough people, especially the old people and patient, who are the potential user of the robotic hand.

BTW, how to get and measure the humans subjective and physiological emotional responses is another problem, which is also very interesting and important.

Author Response

Thank you very much for your review and valuable comments. We appreciate having an opportunity for a point-by-point response. All changes for responding to the comments of all reviewers were shown in blue text.

Reviewer 1 pointed out that the manuscript only focused on the moving speed of the hand and did not consider other factors, such as pressure. Previous research showed that the CT fibers were related to pleasant touch and the response of the CT fibers depended on the stroke speed. Thus, this manuscript focused on the moving speed like other related research methods. It is possible to increase the factors to be considered, but the time to conduct the experiment becomes too long for a participant to make a heavy burden.

So, we add the explanation of our focus in this manuscript in Section 1 and the explanation of the possible future direction in Section 6.

Reviewer 1 pointed out the control of the robot. To unify the robot control for each subject as much as possible, the impedance control provided by the robot manufacturer was used as is. We believe this control is built inside of the robot control loop and is more reliable than the controller that is built outside of the control loop. We described our consideration of robot control in Section 2.2.

Reviewer 1 pointed out the range of the participants. We completely agree that it is important to participate in a diverse range of participants. The main argument of this paper is to show whether robots can provide pleasant touch strokes in a manner similar to humans. We would like to clarify this argument by reducing the complicated individual factors by limiting the age range. We described the selection of the participants in Section 3.1 and the future direction of the participants in Section 6.

Reviewer 1 pointed out the method for measuring the human’s subjective and physiological emotional responses. We carefully followed the measurement method that the other related research used as described in Section 3.4 and did our best for the measurement.

Reviewer 2 Report

Thank you very much for the opportunity to review this very interesting article.

In their work the authors compare the effects of a stroke touch from a robot and a human.

Overall, it is a well-written, clear and concise article, investigating a novel topic. The methodology is sound and the results and conclusions are well-supported by the data.

Author Response

Thank you very much for your review and positive evaluation of our paper.

Reviewer 3 Report

1. Well written in perfect English. Use of jargon was kept to a minimum. Should be readily understandable to specialists and accessible to be the general reader too. 

2. For future research, you cannot ignore the psychological factor. In the future, people will have robot caregivers that they know are robots. Their prejudices will come into play. How do people react when they know they being touched by a human and when they know they are being touched by a robot?

3. I also think the research should be repeated with all female participants.

Author Response

Thank you very much for your review and valuable comments. We appreciate having an opportunity for a point-by-point response. All changes for responding to the comments of all reviewers were shown in blue text.

Reviewer 3 pointed out two future research directions: the psychological factors caused by seeing a robot stroking and female participants. In this study, it is difficult to investigate the psychological factor due to the experimental condition that the participant avoids perceiving the existence of the human agent. We added the explanation of the investigation of the psychological factor by describing it as one of our future works.

We also described the latter as one of our future works in Section 6. In this paper, we attempted to clarify the hypothesis described in Section 1 by reducing the complicated individual factors.

Round 2

Reviewer 1 Report

The manuscript has been revised according to the comments or given some explanations about the comments. It can be published.